# Distinct Cerebrospinal Fluid Lipid Signature in Patients with Subarachnoid Hemorrhage-Induced Hydrocephalus

**DOI:** 10.3390/biomedicines11092360

**Published:** 2023-08-23

**Authors:** Trine L. Toft-Bertelsen, Søren Norge Andreassen, Nina Rostgaard, Markus Harboe Olsen, Nicolas H. Norager, Tenna Capion, Marianne Juhler, Nanna MacAulay

**Affiliations:** 1Department of Neuroscience, University of Copenhagen, 2200 Copenhagen, Denmark; trineto@sund.ku.dk (T.L.T.-B.);; 2Department of Neurosurgery, Neuroscience Centre, Copenhagen University Hospital—Rigshospitalet, 2100 Copenhagen, Denmark; 3Department of Neuroanaesthesiology, Neuroscience Centre, Copenhagen University Hospital—Rigshospitalet, 2100 Copenhagen, Denmark; 4Department of Clinical Medicine, University of Copenhagen, 2200 Copenhagen, Denmark

**Keywords:** cerebrospinal fluid, lipidomics, mass spectrometry, posthemorrhagic hydrocephalus, SAH

## Abstract

Patients with subarachnoid hemorrhage (SAH) may develop posthemorrhagic hydrocephalus (PHH), which is treated with surgical cerebrospinal fluid (CSF) diversion. This diversion is associated with risk of infection and shunt failure. Biomarkers for PHH etiology, CSF dynamics disturbances, and potentially subsequent shunt dependency are therefore in demand. With the recent demonstration of lipid-mediated CSF hypersecretion contributing to PHH, exploration of the CSF lipid signature in relation to brain pathology is of interest. Despite being a relatively new addition to the omic’s landscape, lipidomics are increasingly recognized as a tool for biomarker identification, as they provide a comprehensive overview of lipid profiles in biological systems. We here employ an untargeted mass spectroscopy-based platform and reveal the complete lipid profile of cisternal CSF from healthy control subjects and demonstrate its bimodal fluctuation with age. Various classes of lipids, in addition to select individual lipids, were elevated in the ventricular CSF obtained from patients with SAH during placement of an external ventricular drain. The lipidomic signature of the CSF in the patients with SAH suggests dysregulation of the lipids in the CSF in this patient group. Our data thereby reveal possible biomarkers present in a brain pathology with a hemorrhagic event, some of which could be potential future biomarkers for hypersecretion contributing to ventriculomegaly and thus pharmacological targets for pathologies involving disturbed CSF dynamics.

## 1. Introduction

Non-traumatic subarachnoid hemorrhage (SAH) accounts for 5% of all strokes [1,2,3] and mostly occurs from an aneurysmal rupture [4,5,6,7]. A common complication in patients with SAH is posthemorrhagic hydrocephalus (PHH) [8,9], a condition with accumulation of cerebrospinal fluid (CSF) within the ventricles that causes hydrocephalus and elevation of the intracranial pressure (ICP). Currently, the initial treatment relies entirely on pressure reduction by drainage of the excess fluid from the ventricles via surgical CSF diversion using an external ventricular drain and, subsequently, for those who develop chronic hydrocephalus, also a ventriculoperitoneal shunt implantation [10,11,12], as no sufficiently efficient pharmacological options exist due to the elusive molecular nature of the CSF secretion apparatus and its regulatory properties.

During the last decade there has been a sustained effort to identify various types of acute brain injury-associated biomarkers for indication of cerebral damage [13,14,15], and several studies have been conducted to identify biomarkers for shunt dependency in patients with PHH following SAH, without succeeding in finding good predictors [16,17,18,19]. Biomarkers that convincingly predict PHH etiology and severity, shunt dependency, and patient outcome, are therefore in demand. Lipids are underrepresented in these efforts, although the brain is the organ with the highest content of lipids, i.e., fatty acids, phospholipids, sphingolipids, glycerolipids, and sterols [20]. Lipids are a major class of cellular components, which shows a wide diversity and presents with a variety of structural and signaling roles implicated in the regulation of a range of vital cellular processes such as cell differentiation, proliferation, survival, and apoptosis [21]. Lipids can thus be anticipated to be strictly regulated. Notably, dysregulation of various lipids has been observed in patients with SAH/PHH [22] and in a SAH animal model thereof [23]. Of these, lysophosphatidic acid (LPA) was found upregulated in patients and rats with PHH [23,24] and traumatic brain injury (TBI) [25,26,27,28,29,30]. This serum lipid enters the ventricular system with the hemorrhagic event and promotes the CSF hypersecretion contributing to ventriculomegaly and PHH formation, via its action as a modulator of the CSF secretion machinery [23].

The lipid profile of ventricular CSF in the group of patients with SAH remains unresolved and, therefore, per extension, also the potential presence of other lipids that may serve modulatory roles in the CSF secretion apparatus. The development of lipidomics, (i.e., the study of lipid classes, lipid networks and pathways) has allowed insights into the complexity of lipid profiles and their dynamics. Such analysis provides a complete lipid profile of a given compartment and is thus suitable for exploring lipid homeostasis in brain diseases. Liquid chromatography coupled to tandem mass spectrometry (LC-MS/MS), as employed in this study, is an effective analytic platform for untargeted lipidomics with high accuracy and resolution. Here, we employed this technique to determine the lipid profile in CSF obtained from healthy individuals undergoing prophylactic vascular clipping of an unruptured aneurysm (henceforth referred to as controls) and patients with SAH who subsequently developed chronic hydrocephalus in order to identify a specific lipid signature in patients with SAH-associated hydrocephalus that may modulate the CSF dynamics and potentially contribute to the ventriculomegaly. We successfully demonstrate dysregulated CSF lipid levels in patients with SAH-induced PHH.

## 2. Materials and Methods

### 2.1. Patients

Ventricular CSF samples were collected between June 2019 and September 2021 from 13 patients with SAH admitted and treated at the Department of Neurosurgery at Rigshospitalet, Copenhagen, Denmark (Table 1). All patients had acute hydrocephalus as verified with CT scanning. The ventricular CSF was sampled within 24 h of ictus (*n* = 8) or as soon as possible hereafter (*n* = 5) (range: 3–74 h). Only patients with no signs of neuroinfection at admission or during their treatment were selected to ensure that measured CSF parameters were not affected by neuroinfections requiring antibiotic treatment. Patient samples were randomly chosen from our biobank to fit the following criterion: all enrolled patients with SAH received a permanent ventriculo-peritoneal shunt because of continued need for CSF diversion (development of PHH). Patients who had milder PHH, and thus did not need a shunt, were therefore not included in this study. As control subjects, 11 patients undergoing preventive surgery for unruptured aneurysms (vascular clipping) were enrolled, and CSF was collected from the basal cisterns during surgery prior to clipping of the aneurysm (Table 1). There was no statistical significance between the SAH group and the control subjects with regard to the parameters listed in Table 1. The CSF sampling for analytical purposes was carried out upon placement of the EVD that was inserted on clinical indication and the CSF therefore did not reside in the EVD prior to being centrifuged at 2000× *g* for 10 min at 4 °C within 2 h from collection, and subsequently stored at −80 °C. Written informed consent was obtained from all patients or next of kin depending on the capacity of the patients. The study was approved by the Ethics committee of the Capital Region of Denmark (H-19001474/69197 and H-17011472). CSF samples obtained from patients within the same cohort had previously been analyzed for inflammatory markers [31,32]. 

### 2.2. Liquid Chromatography/Mass Spectrometry (LC-MS) Analysis

CSF was transferred into a Spin-X^®^ filter with extraction solvent, and centrifuged at 1400 rpm at 5 °C for 2 min. Following this, the sample was transferred into a high recovery HPLC vial with eluent mix and capped. Liquid chromatography-mass spectroscopy (using two ultra-high-pressure liquid chromatography instruments coupled with Thermo Scientific Vanquish LC (Lund, Sweden) coupled to Thermo Q Exactive HF mass spectrometers) was used as untargeted analysis of the CSF samples and conducted by MS-Omics, Denmark. Ionization was performed in positive and negative ionization mode using an electrospray ionization interface. The chromatographic separation of lipids was carried out on a Waters^®^ (Milford, MA, USA) ACQUITY Charged Surface Hybrid (CSH™) C18 column (2.1 mm × 100 mm, 1.7 µm). The column was thermostated at 55 °C. The mobile phases consisted of (A) acetonitrile/water (60:40) and (B) isopropanol/acetonitrile (90:10), both with 10 mM ammonium formate and 0.1% formic acid. Lipids were eluted in a two-step gradient by increasing B in A from 40 to 99% over 18 min. Flow rate was 0.4 mL/min. The MS was operated with the following settings: *m*/*z* range: 200–1500; AGC target: 1 × 10^6^; resolution: 120,000; collision energy: 30. Compounds were extracted based on features (a peak characterized by one mass and one retention time), and additional information (e.g., isotope pattern, accurate mass, and molecular formula). The retention time of compounds within the same lipid class is estimated using the relation between retention time and the chain length and number of double bonds. Three levels of annotation were used (level 1–3) with level 1 representing the most confident identifications (based on accurate mass, tandem mass spectrometry spectra (MSMS) and estimated retention time), level 2, based on two of these sets of information, is divided into two sublevels; accurate mass and estimated retention, and level 3 is based on library searches using the accurate mass and elemental composition alone. In this study, level 3 annotations were not included. As a quality control, a mixed pool sample was created by taking a small aliquot from each sample to be analyzed with regular intervals through the sequence. Splash lipidomix (Avanti 330707), a single-vial prepared lipidomic analytical standard for human plasma lipids, was diluted 1:50 as prescribed by the manufacturer and used as internal standards for QC check (retention time, intensity, and mass accuracy) to ensure that all samples were analyzed correctly. The mixture components were 15:0–18:1(d7)PC (150.6 μg/mL), 15:0–18:1(d7)PE (5.3 μg/mL), 15:0–18:1(d7)PS (3.9 μg/mL), 15:0–18:1(d7)PG (26.7 μg/mL), 15:0–18:1(d7)PI (8.5 μg/mL), 15:0–18:1(d7)PA (6.9 μg/mL), 18:1(d7)LPC (23.8 μg/mL), 18:1(d7)LPE (4.9 μg/mL), 18:1(d7)cholesterol (329.1 μg/mL), 18:1(d7)MG (1.8 μg/mL), 15:0–18:1(d7)DG (8.8 μg/mL), 15:0–18:1(d7)-15:0 TG (52.8 μg/mL), 18:1(d9)SM (29.6 μg/mL), cholesterol(d7) (98.4 μg/mL). Data were processed using Compound Discoverer 3.0 (ThermoFisher Scientific, Waltham, MA, USA). 

### 2.3. Bioinformatics and Statistical Analysis

A Smirnov−Grubbs test (two sided, α = 0.05) was performed for all detected compounds (358) within each patient group (control, *n* = 11; SAH, *n* = 13) and the number of outliers was counted for each sample. If a patient sample presented more than 20% outliers, this patient sample was excluded from the rest of the analysis (excluded samples: control = 2, SAH = 1). All 358 compounds were manually curated into 35 main classes of lipids, non-biological lipid compounds, and non-classified (‘others’), of which the latter two groups were excluded from the analyses. Fourteen groups, all containing less than four lipids (<1% of the total detected compounds), were assembled into one group (‘small group collection’). The resulting 17 groups with 244 compounds were employed for the analyses. An enrichment plot of the number of lipids detected within each group was generated for the 17 groups counting all compounds assigned to each group. To obtain the total lipid abundance for each group, the lipids were normalized based on the geomean of the quality control samples (*n* = 15). An enrichment plot was generated for normalized mean abundance by summing the normalized values of lipids in each group upon which the mean of all controls or patients with SAH was calculated. Principal component analysis (PCA) [33] plots were employed to detect separation of the two groups and were generated with the normalized lipid values for each sample and either grouped for sex, age, and SAH/control or displayed within each lipid group. The line plots were generated with normalized mean abundance for each lipid group as a function of the age of the subjects. All quantifications in the volcano plot and the bar plots were generated using normalized data without outliers (two-sided Smirnov−Grubbs test, α = 0.05) and excluding lipids with a descriptive power of less than 2.5 (the ratio between the standard deviations of all experimental samples and the quality control samples). Statistical tests were conducted with Welch’s *t*-test followed by the Benjamini−Hochberg method (with an adjusted *p* value < 0.1 (false discovery rate, FDR, of 10%) [34,35]. All scripts for the data analysis can be found at: https://github.com/Sorennorge/MacAulayLab-SAH-Metabolomics (accessed on 8 August 2023).

## 3. Results

### 3.1. CSF Lipid Profiles in Control Subjects

To delineate the lipid composition in the cisternal CSF obtained from control subjects (Table 1) undergoing clipping of an unruptured aneurysm, we obtained CSF from the surgically accessed cisterns. These samples were analyzed with non-targeted lipidomics, which detected 357 representative compounds (Appendix A), of which 244 were classified as lipid compounds assigned to 17 groups according to their main class (Figure 1A). Groups with less than four lipids (<1%; see Appendix A) were concatenated into a single group termed the ‘small group collection’. Although not classified as biological lipids, various sugars, amino acids, vitamins, medics employed in the clinic, and non-classified detected molecules appeared in the analysis (Appendix A), but were omitted from the analyses performed in the study. The distribution profile revealed 43 different triacylglycerols (17.6%) and 43 different phosphatidylcholines (17.6%), which then represented the lipid classes with most representatives. Sphingomyelines were the second most represented lipid class (24 different lipids; 9.8%) followed by fatty acids and plasmenylphosphatidylcholines (both 22 different lipids, 9.0%), Figure 1A. To determine the representation of lipids within each group, we quantified the abundance of lipids within each lipid group (Figure 1B), which demonstrated that phosphatidylcholine was the lipid class with the highest lipid abundance present (20.4%), followed by triacylglycerols (15.2%), sphingomyelines (13.2%), and plasmenylphosphatidylcholines (13.1%).

### 3.2. No Age- and Sex-Dependent CSF Lipid Distribution in Control Subjects

To reveal a potential sex-dependent CSF lipid distribution in control individuals, we employed a principal component analysis (PCA). The comparison of overall lipid profiles of each sample revealed a similar lipid distribution in CSF obtained in males and females with no distinct grouping of sexes (Figure 2A), which was replicated with independent PCA analysis of the individual main classes of lipids (Appendix A). To determine the age-associated CSF lipid distribution, the subjects were initially divided at the median age (60 years old), which led to a group of five individuals ≤ 60 years and one with six individuals > 60 years. Comparison of the total lipid profiles from these two groups revealed no overall age-dependent lipid grouping (Figure 2B), as was also the case with independent PCA analysis of the individual main lipid classes in the two age groups (Appendix A). Despite the lack of an overall distinct lipid distribution pattern within the two age groups, we determined the lipid group abundance fluctuation as a function of age from the youngest (40 years old) to the oldest (77 years old) subject. Within this sample set, the lipid content appeared to fluctuate in a bimodal manner with a peak around the mid-50s and another, smaller one, around the mid-70s, with low points around the 40-year and 60-year marks (Figure 2C). Although the overall pattern was similar for the lipid groups, some displayed more obvious fluctuations (i.e., lysophosphatidylcholines, phosphatidylcholines, plasmenylphosphatidylcholines, sphingomyelines, triacylglycerols) and some barely, if at all (i.e., monoacrylglycerols, fatty acids, phosphatidic acids), see Appendix A for individual lipid distribution. 

### 3.3. CSF Lipid Profile in Patients with SAH

To map the lipid composition in patients with SAH (Table 1), we employed ventricular CSF samples obtained during the placement of an external ventricular drain upon admission to the neurosurgical department. The 244 detected ventricular CSF lipid compounds in this cohort were organized into 17 groups according to their main class as presented for the control subjects. Detected non-biological lipid compounds (Appendix A) were omitted from the analysis. Quantification of lipid abundance within each lipid group demonstrated that phosphatidylcholine was the lipid class with the highest lipid abundance in the sampled CSF (19.0%), followed by triacylglycerols (17.8%), sphingomyelines (9.4%), and plasmenylphosphatidylcholines (8.4%), Figure 3A. PCA plots revealed no age- or sex-dependent distribution of these lipids in the patients whether taken across the entire lipid composition (Figure 3B,C) or when divided into individual main classes (Appendix A).

### 3.4. Dysregulated CSF Lipid Levels in Patients with SAH

To determine whether patients with SAH display a distinct CSF lipid signature, we analyzed the lipid compounds within the two groups of subjects with a PCA plot. The results demonstrated variations in the total CSF lipid profile, which separated the patients with SAH from the control subjects (Figure 4A), although with no difference in the overall lipid concentration (mean abundance; 18.3 ± 3.7 in patients with SAH vs. 15.2 ± 2.3 in control subjects, *p* = 0.49). To identify the individual lipid classes that drive the diversion of the overall SAH lipid profile from the control subjects, we compared the total abundance of each lipid class between the two groups. On a lipid class basis, the abundance of fatty acids, phosphatidic acids, plasmenylphosphatidylethanolamines, and monoglycerols was significantly elevated in the CSF from patients with SAH (Figure 4B). PCA plots of the individual classes of lipids, in addition, revealed separation of amides and lysophosphatidylcholines (and to a lesser extent phosphatidylethanolamines) in patients with SAH compared to control subjects (Appendix A), which suggests overall dysregulation of select lipid classes in CSF from patients with SAH. To identify individual dysregulated lipids in patients with SAH compared to control subjects, we subsequently arranged the data in a Volcano plot, which revealed seven lipids significantly upregulated in CSF in patients with SAH, when correcting for multiple testing (Figure 4C). These dysregulated lipids fell into four lipid classes, with amide C18 and arachidonoyl amide from the amide class, PC 30:0, PC 31:0, and PC 32:0; PC (16:0/16:0) from the phosphatidylcholine class, plasmenyl-PE 43:6 from the plasmenylphosphatidylethanolamine class, and plasmenyl-PE 36:1 (18:0/18:1). A cluster of lipids containing fatty acids, lysophosphatidylcholines, monoacrylglycerols, phosphatidic acids, and the small group collection was detected as highly elevated (>2.5 log2FC; 5–16-fold increase) in patients with SAH, but only significantly so without correction for multiple testing (Figure 4C). This finding prompted us to analyze the lipid content within each lipid class in isolation, where we revealed dysregulation of lipids within ten of the lipid classes (Figure 5; Appendix A), representing the drivers of lipid profile separation in patients with SAH compared to control subjects. Most of these displayed elevation of several lipids within the class (i.e., phosphatidic acids, phosphatidylcholines, lysophosphatidylcholines, fatty acids, and monoacrylglycerols), whereas some of the lipids from the class of amides displayed downregulation in CSF from patients with SAH compared to control subjects. Taken together, a range of CSF lipids appeared dysregulated in patients with SAH, mainly evident as an increase in the lipid abundance.

## 4. Discussion

Here we report CSF dysregulated lipid levels in a group of patients with SAH as compared to CSF obtained from control subjects. We detected the presence of 357 compounds, of which 244 were categorized into 17 main lipid classes with triacylglycerols and phosphatidylcholines as the dominant lipids in this compartment. The lipid distribution was independent of sex, whether categorized overall or according to the different lipid classes. This finding contrasts with a previous report that indicated some forms of sphingomyelins (e.g., SMd42:1) to associate with the female sex [36], although this conclusion was based on CSF-plasma correlations of lipids [36] and not on absolute lipid abundance in the CSF. Upon division of the individuals into two groups according to the median age, we observed a similar lipid distribution in their CSF. A previous study demonstrated an age-dependent lipid profile of CSF of plasmalogens, i.e., plasmenylphosphatidylethanolamines, fatty acids, and phosphatidylcholines in control subjects [36]. However, these samples were obtained from the lumbar CSF compartment and arranged with a median split at 40 years [20]. This discrepancy aligned well with our demonstration of a bimodal fluctuation in CSF lipid level as a function of age, with the larger peak occurring around that very age group (~50 years). Notably, most of the lipid groups appear to fluctuate with age whereas others did not (fatty acids, monoacrylglycerols, phosphatidic acids). One would be required to take such age-dependent fluctuations in CSF lipids into account with potential future employment of these as biomarkers for SAH.

Patients with SAH demonstrated a distinct CSF lipid signature with upregulation of several classes of lipids, i.e., phosphatidylcholines, plasmenylphosphatidylethanolamines, phosphatidic acids, fatty acids, and monoacrylglycerols. In addition, a cluster of lipids were detected as highly elevated in patients with SAH, but only significantly so without correction for multiple testing (those lipids fell into the classes of fatty acids, lysophosphatidylcholines, monoacrylglycerols, phosphatidic acids, and the small group collection). With analysis of each lipid class in isolation, a range of individual lipids stood out as significantly upregulated in the patients with SAH and could potentially serve as biomarkers for SAH progression and PHH formation. Phosphatidylcholines, several of which (PC30:0, PC31:0, and PC32:0 (PC(16:0/16:0)) were upregulated in the CSF from patients with SAH, play critical roles in signaling cascades in cell apoptosis and various inflammatory responses contributing to SAH-induced early brain injury in animal models [37]. In addition, phosphatidylcholines peroxided by hemoglobin contribute to cerebral vasospasm following SAH in animal models [38], as induced by altered lipid metabolism post-SAH [39]. Following SAH, fatty acids, important structural components of neuronal membranes and precursors of signaling molecules [40], were elevated (this study and [41]), and suggested to play a potential role in vasospasm that leads to vasoconstriction, thus preventing the flow of blood at a normal rate [41]. Phosphatidylcholines, plasmalogens, and other phospholipids such as glycolipids, are examples of glycerophospholipids, which account for more than 50% of the lipid content of membranes, and 45% of the total dry weight of the brain [42]. Glycerolphospholipids and plasmalogens are major classes in the neural membranes [43] and plasmalogens represent a class of lipids important for brain function [44,45]. We observed an increased level of the plasmalogen plasmenylphosphatidylethanolamine 43:6 in patients with SAH, which contrasts with that observed in patients with Alzheimer’s disease, depression, and bipolar disease [46,47,48]. Our finding of increased levels of the monoacrylglycerol 18:2 (linoleic acid), aligns with a previous study indicating that higher concentrations of particularly polyunsaturated fatty acids are associated with a worse outcome [41]. Other monoacrylglycerols, i.e., oleic, erucic, and conjugated linoleic acids modulate CSF inflammatory markers [49] and could, as such, promote PHH in the group of patients with SAH.

We identified five members of the phosphatidic acid lipid class being upregulated: PA22:0, PA24:0, PA26:0, PA28:0, and PA45:4. Phosphatidic acids are the simplest (diacyl)glycerophospholipids present in cells, a second messenger, and the common intermediate in synthesis of all phospholipids. The levels of phosphatidic acids are often seen to be elevated following activation of inflammatory cells [32]. Lysophosphatidylcholines are increasingly recognized as a key factor associated with cardiovascular and neurodegenerative diseases. These lipids are mainly derived from the turnover of phosphatidylcholine via the action of phospholipase A2 [50] and/or by the transfer of fatty acids to free cholesterol [51]. Amongst the three upregulated lysophosphatidylcholines, we detected an elevation of the lysophosphatidylcholine 18:2, in agreement with a previous study on patients with repetitive mild traumatic brain injury [52]. The dysregulated lipids in the CSF from patients with SAH could thus be potential biomarkers serving as predictors of posthemorrhagic outcome, as has been evident from earlier quantifications of inflammatory markers [31].

The SAH-induced lipid abundance could arise with the entry of blood into the ventricular compartment or as a biological response to the hemorrhagic event. Irrespectively thereof, the CSF lipid composition could contribute to a coupling between a brain hemorrhagic event and the ensuing ventricular enlargement signifying PHH. Removal of a choline group from the lysophosphatidylcholine, by the action of a lysophospholipase, yields LPA, which has earlier been detected in patient CSF following TBI [24] and SAH [23] and in an animal model of intraventricular hemorrhage [23]. Experimental intraventricular administration of LPA to rats mediates ventricular enlargements, third ventricular occlusion, elevated ICP, thinning of cortical layers, and cilia loss along the lateral ventricular walls [53,54], in addition to its direct promotion of CSF hypersecretion [21]. LPA acts as a direct agonist of a choroid plexus ion channel, the transient receptor potential vanilloid 4 (TRPV4) [23,55]. This ion channel increases, via an intracellular signaling cascade, the activity of a choroid plexus transporter involved in CSF secretion [23,56,57], contributing to the ensuing ventriculomegaly signifying PHH [23]. We propose that other lipids dysregulated in CSF from patients with SAH could well serve similar or alternative roles that may directly contribute to the formation of PHH.

### 4.1. Strengths and Limitations

In the current study we employed an untargeted mass spectroscopy-based platform to quantify the lipid content in ventricular and cisternal CSF, which is the first of its kind, contrasting with earlier targeted studies based on CSF obtained from the lumbar CSF compartment [36]. Limitations of the present study include the limited number of patient samples (which was dictated by the limited number of control samples we had access to), especially so for the age-related correlation, the sampling of CSF from the ventricular compartment in the patients with SAH versus from the basal cisterns in the control subjects, and the potential introduction of blood components to the CSF occurring with EVD insertion and/or vascular clipping. The different patient sample sites were dictated by ethical limitations in invasive CSF sampling but could influence our results if the CSF composition differs between these locations. In addition, with the advantage of untargeted quantification of a large number of lipids in the CSF comes the statistical requirement for correction for multiple sampling. Such a correction may cause false negatives, in the sense that a single lipid, tested singularly by a benchtop assay, could serve as a future predictor of SAH outcome or shunt dependency, but be lost in an unbiased quantification, as here employed. Of note, we therefore—with caution—included additional testing of groups of lipids in our analysis that still displayed a robust fold change between the CFS samples from our two test groups. The majority of CSF samplings were done within 24 h after ictus with no apparent time-dependent lipid distribution in one of our previous studies with determination of a single lipid in a similar patient group [23]. However, some patients were in the so-called early brain injury period upon CSF sampling, while others were closer to that vasospasm window. No special consideration was given to concurrent or previous use of statins, insulin, or other drugs that could affect the lipid profile in patients with SAH as this information was not accessible.

### 4.2. Clinical Relevance

Biomarkers which reliably reflect CSF homeostasis following SAH would be extremely useful for clinical assessment and decision making. If such biomarkers can be used to distinguish between individuals with only a temporary need for CSF diversion and those with a long-term or permanent need, EVD discontinuation in the former group vs. shunt insertion in the latter group could be decided earlier, making it possible for patients to leave intensive care unit environments and begin rehabilitation. Biomarkers reflecting the extent of brain tissue damage would be useful to assess functional prognosis and rehabilitation potential and might also assist in distinguishing exhausted rehabilitation potential due to the primary brain damage from shunt dysfunction.

## 5. Conclusions

In conclusion, we demonstrate that, of the 357 detected compounds, 244 lipids assigned to 17 different classes made up the lipid profile of the control subjects. The lipid distribution is not sex dependent but does fluctuate with age. Our results revealed a CSF lipid signature for patients with SAH, which encourages future studies aimed at elucidating if any of the identified dysregulated lipids induces pathophysiological changes in the brain fluid dynamics occurring with SAH. Such studies could aim to identify novel agonists of the molecular machinery underlying lipid-induced CSF hypersecretion, and thus novel lipid biomarkers with the potential to be employed in predicting the shunt dependency and/or outcome of SAH.

## Figures and Tables

**Figure 1 biomedicines-11-02360-f001:**
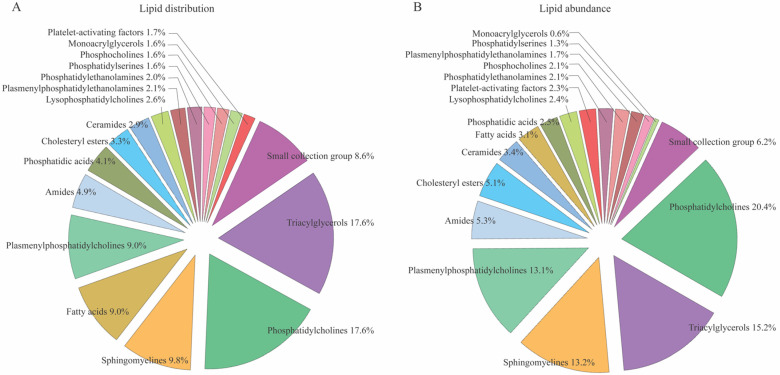
CSF lipidomics in control subjects. (**A**) Detected CSF lipids (244 compounds) were distributed according to classes (17 classes), based on enrichment analysis. Fourteen groups contained less than four lipids (<1% of the total detected compounds) and were assembled into one group (‘small collection group’). (**B**) Lipids were normalized based on the geomean of the quality control samples (*n* = 15) to obtain the total lipid abundance for each group, and an enrichment plot was generated by summing the normalized values of lipids in each class upon which the mean of all controls was calculated. Data are based on cisternal CSF from control subjects (*n* = 11).

**Figure 2 biomedicines-11-02360-f002:**
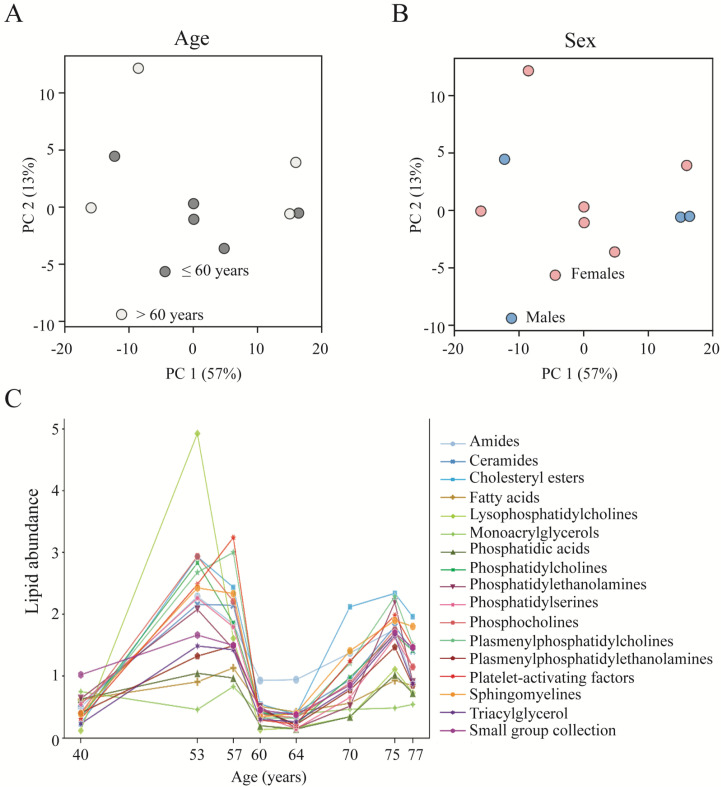
Subject background and CSF lipid distribution in control subjects. (**A**) The normalized total lipid values (with a descriptive power above 2.5) were grouped for age initially divided at the median age (light gray spheres: ≤60 years; dark gray spheres: >60 years) or sex (**B**) (blue: male; pink: female) and plotted with respect to their first and second principal components (PC1 and PC2). (**C**) Total lipid abundance fluctuation was generated based on normalized mean abundance for each lipid group as a function of the age of the control subjects and plotted as a line plot from the youngest (40 years old) to the oldest (77 years old) control subject. Data are based on cisternal CSF from control subjects (*n* = 11; note that the spheres indicating ages 57 and 77 years are based on two subjects each).

**Figure 3 biomedicines-11-02360-f003:**
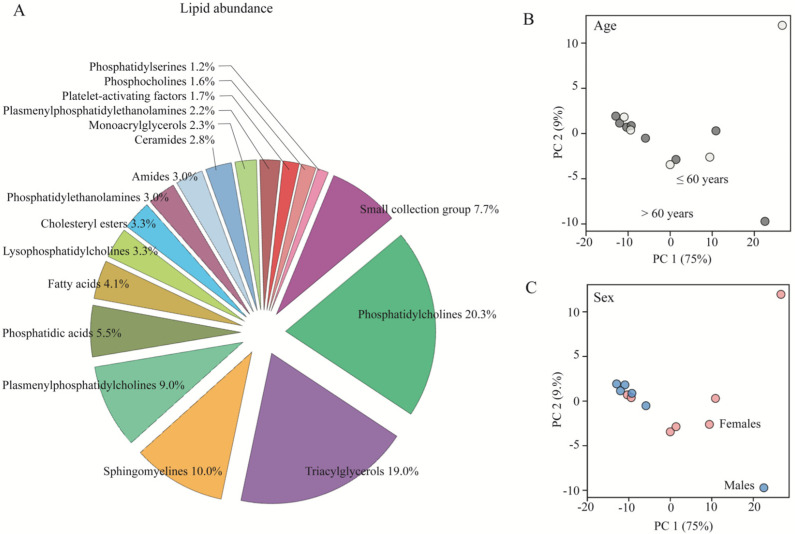
CSF lipid profile in patients with SAH. (**A**) The total abundance for each CSF lipid class was obtained with normalization based on the geomean of the quality control samples (*n* = 15). An enrichment plot was generated based on the summing of normalized values of lipids in each class from patients with SAH. (**B**) The normalized lipid values (with a descriptive power above 2.5) were grouped for age (light gray spheres: ≤60 years; dark gray spheres: >60 years) or sex (**C**) (blue: male; pink: female) and plotted with respect to their first and second principal components (PC1 and PC2). Data are based on ventricular CSF from patients with SAH (*n* = 13).

**Figure 4 biomedicines-11-02360-f004:**
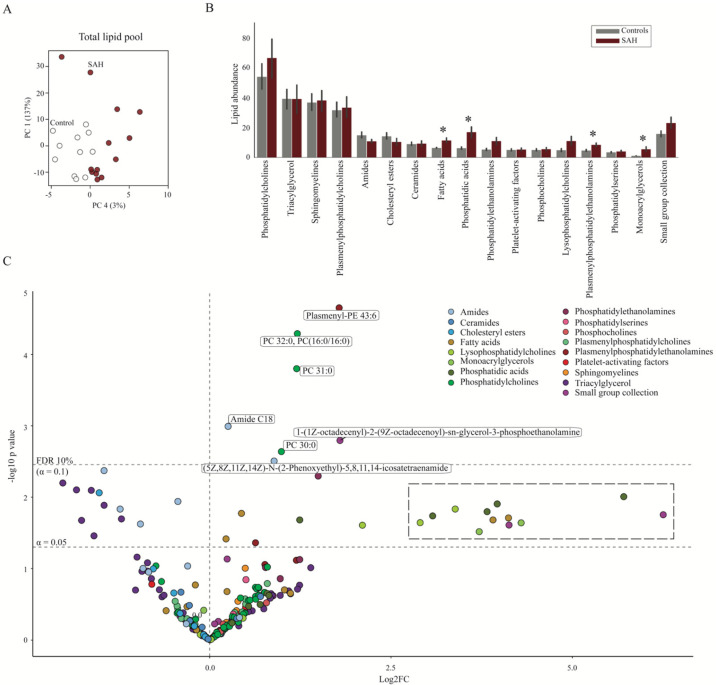
Dysregulated CSF lipid levels in patients with SAH. (**A**) The normalized total lipid values (with a descriptive power above 2.5) were grouped for control subjects and patients with SAH (white spheres: control subjects; red: SAH) and plotted with respect to their first and fourth principal components (PC1 and PC4). (**B**) The total abundance of each lipid class in CSF from control subjects and patients with SAH was plotted as bars using normalized data. (**C**) Volcano plot of normalized individual CSF lipids identified with the fold change (log2 transformed) between control subjects and patients with SAH. Upper dashed line: adjusted *p* value; indicates cut-off for significance. Lower dashed line: *p* value < 0.05. A cluster of lipids that were detected as highly elevated in patients with SAH, although not reaching the adjusted significance level, are highlighted by a dashed line box. Statistical evaluation with Welch’s *t*-test followed by the Benjamini−Hochberg method (with an adjusted *p* value < 0.1 (false discovery rate, FDR, of 10%). * *p* < 0.05. FC: fold change.

**Figure 5 biomedicines-11-02360-f005:**
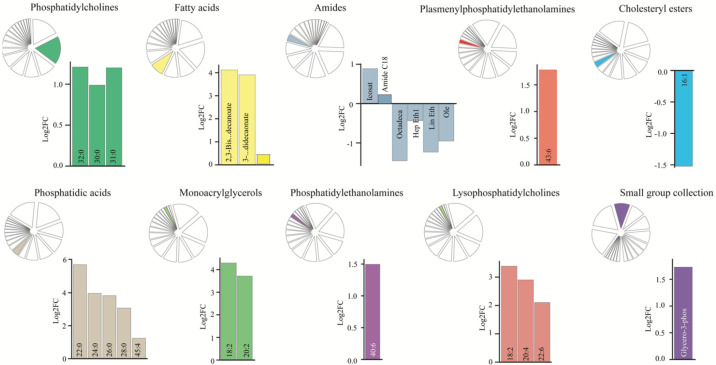
Isolated lipid class analysis. The lipid content within each lipid class in isolation between control subjects and patients with SAH are plotted as bars using normalized data (without outliers; two-sided Smirnov−Grubbs test, α = 0.05). Lipids with a descriptive power of less than 2.5 were excluded from the analysis. FC: fold change.

**Table 1 biomedicines-11-02360-t001:** Clinical characteristics of the study cohort. *n*: number of included individuals; F: female; M: male; SAH: subarachnoid hemorrhage; BMI: body mass index. Statistical analysis of parameters within the two cohorts was carried out with an unpaired Student’s *t*-test.

Study Cohort	SAH	Control (Unruptured Aneurysm)	Statistics
*n*	13	11	
Age (years), median (range)	62 (44–71)	64 (40–77)	*p* = 0.66
Sex (F/M)	7F/6M	7F/4M	
BMI (kg/m^2^), median (range)	26 (21.8–32.1)	25 (18.9–35.6)	*p* = 0.57

## Data Availability

The dataset supporting the conclusion of this article is available in the MacAulay Lab database repository, https://github.com/Sorennorge/MacAulayLab-SAH-Metabolomics, accessed on 8 August 2023.

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
