# Peer review of "Distinct Cerebrospinal Fluid Lipid Signature in Patients with Subarachnoid Hemorrhage-Induced Hydrocephalus"

_biomedicines, 2023, doi:10.3390/biomedicines11092360_

Round 1
Reviewer 1 Report
Dear the Editor
Toft-Bertelsen TL et al reported the lipidomics profile in CSF from individuals with subarchnoid hemorrhage (SAH)-induced hydrocephalus. As shown in Table 1, both SAH and control subject had a similar demography with an averaged age of 61 years old for SAH and 64 years old for Control. In CSF, these authors reported that TG, PC, free fatty acids, SM, and plasmenylphosphosphatidylcholine and plasmenylethanolamine (collectively plasmalogens) were major lipid components. Overall, this manuscript seemed organized well.
Major concerns:
1) In Fig. 4C, please comment why PC31:0 is elevated in SAH. It would be interesting to discuss why 17:0 or 14:0, two free fatty acids of PC31:0, were elevated. Because 17:0 is not biosynthesized in humans, thus please discuss the origin of this.
2) Is there any alteration in cholesterol or its derivative? It is known that cholesterol can be biosynthesized in the brain, therefore it does not require penetration through blood-brain barrier.
3) Apart from alteration of lipid profiles, is there any changes in the concentration of lipids?
Minor concerns:
1) In Method section (p3), please describe the name of lipid, its concentration, and commercial source used for internal standard.
Author Response
Reviewer 1
Toft-Bertelsen TL et al. reported the lipidomics profile in CSF from individuals with subarchnoid hemorrhage (SAH)-induced hydrocephalus. As shown in Table 1, both SAH and control subject had a similar demography with an averaged age of 61 years old for SAH and 64 years old for Control. In CSF, these authors reported that TG, PC, free fatty acids, SM, and plasmenylphosphosphatidylcholine and plasmenylethanolamine (collectively plasmalogens) were major lipid components. Overall, this manuscript seemed organized well.
Answer: We thank the reviewer for the positive comments.
Major concerns:
1) In Fig. 4C, please comment why PC31:0 is elevated in SAH. It would be interesting to discuss why 17:0 or 14:0, two free fatty acids of PC31:0, were elevated. Because 17:0 is not biosynthesized in humans, thus please discuss the origin of this.
Answer: As the reviewer states, FA17:0 is not biosynthetized in humans. Accordingly, it does not show up in our analysis. It is LysoPC17:0 (and also FA 14:0, as the reviewer mentions), but neither is elevated in our analysis.
2) Is there any alteration in cholesterol or its derivative? It is known that cholesterol can be biosynthesized in the brain, therefore it does not require penetration through blood-brain barrier.
Answer: Some, if not all, the lipids detected as elevated in SAH may be expected to enter with the hemorrhagic event and therefore will not require BBB permeation to enter, we suspect. However, we detected a downregulation of the cholesterol derivative cholesteryl ester 16:3, see Fig.5.
3) Apart from alteration of lipid profiles, is there any changes in the concentration of lipids?
Answer: We did not detect a change in the overall lipid concentrations in the patients with SAH compared to the control group. The lipid concentration and statistics are included in the revised version, page 9, line 230.
Minor concerns:
4) In Method section (p3), please describe the name of lipid, its concentration, and commercial source used for internal standard.
Answer: We employed Splash lipidomix (Avanti 330707) as internal standard for QC check (retention time, intensity, and mass accuracy), a single-vial prepared lipidomic analytical standard for human plasma lipids (diluted 1:50 as prescribed by the manufacturer). We have included additional information regarding the lipid content of the internal standard in the revised manuscript, page 3, line 125-128).
Reviewer 2 Report
The Danish authors' manuscript is prepared quite carefully and is of potentially practical importance in diagnosing brain pathologies.
Here are the comments on the manuscript:
1. title - I'm not sure if using an abbreviation, even as widely known as CSF, is correct.
2. introduction - consists of one paragraph, which does not make it easier to read its content. So I propose to break the introduction into smaller parts. Furthermore, I have no criticisms of this part of the manuscript.
3. Table 1 - please compare the values from this table, namely were there statistically significant differences between the parameters?
4. PCA - please provide literature based on which it can be assumed that this chemometric method is useful in rather medical research. My suggestion is as follows: doi: 10.3109/14767058.2012.735999.
5. results - clearly presented, my minor remark concerns only the poor quality of figures.
6. discussion - I congratulate the authors for drawing attention to two issues that are often overlooked by others - the clinical significance of the research and the limitations of the study.
Overall, the manuscript is very well written and provides new data of potentially practical importance in the clinic.
Author Response
Reviewer 2
The Danish authors' manuscript is prepared quite carefully and is of potentially practical importance in diagnosing brain pathologies.
Answer: We thank the reviewer for the positive comments.
Here are the comments on the manuscript:
- title - I'm not sure if using an abbreviation, even as widely known as CSF, is correct.
Answer: Good point. We have changed the title accordingly and replaced ‘CSF’ with ‘cerebrospinal fluid’.
- introduction - consists of one paragraph, which does not make it easier to read its content. So I propose to break the introduction into smaller parts. Furthermore, I have no criticisms of this part of the manuscript.
Answer: We agree with the reviewer and have introduced two breaks in the revised introduction.
- Table 1 - please compare the values from this table, namely were there statistically significant differences between the parameters?
Answer: There was no statistical difference between the two groups with regard to the parameters listed in Table 1. This information is now included in Table 1 and mentioned in the revised method section (page 2 line 86).
- PCA - please provide literature based on which it can be assumed that this chemometric method is useful in rather medical research. My suggestion is as follows: doi: 10.3109/14767058.2012.735999.
Answer: It is our impression that PCA plots serve to compare groups irrespective of the type of data contained in the plots, so also highly relevant for clinical data, as demonstrated in the suggested article. We have included this information in the revised method section (page 3 line 142).
- results - clearly presented, my minor remark concerns only the poor quality of figures.
Answer: We apologize. In the effort to conform to the template setup, the figures were compressed. The revised version of the manuscript contains figures with other dimensions, and hope that they are clearer now (and that we will be permitted to submit high-resolution figures prior to publication).
- discussion - I congratulate the authors for drawing attention to two issues that are often overlooked by others - the clinical significance of the research and the limitations of the study.
Answer: We agree and thank the reviewer for specifically acknowledging these two sections in our manuscript.
Overall, the manuscript is very well written and provides new data of potentially practical importance in the clinic.
Round 2
Reviewer 1 Report
Dear the Editor,
All raised concerns by this Reviewer were properly addressed by the Authors by point-by-point manner.
Author Response
We thank the reviewer for the positive comment and highly competent input.
Reviewer 2 Report
The authors have addressed my remarks in a thorough and satisfactory fashion.
Author Response

(The authors gave the same response as above.)
